# Nutritional Intake and Training Load of Professional Female Football Players during a Mid-Season Microcycle

**DOI:** 10.3390/nu14102149

**Published:** 2022-05-21

**Authors:** César Leão, António Pedro Mendes, Catarina Custódio, Mafalda Ng, Nuno Ribeiro, Nuno Loureiro, João Pedro Araújo, José Afonso, Sílvia Rocha-Rodrigues, Francisco Tavares

**Affiliations:** 1Escola Superior de Desporto e Lazer, Instituto Politécnico de Viana do Castelo, Rua Escola Industrial e Comercial de Nun’Alvares, 4900-347 Viana do Castelo, Portugal; cleao@esdl.ipvc.pt; 2Medical and Performance Department, Sporting Clube de Portugal, Estrada da Malhada de Meias, Barroca d’Alva, 2890-529 Lisboa, Portugal; acmendes@sporting.pt (A.P.M.); catarinacustodio5@gmail.com (C.C.); mcdsilva@sporting.pt (M.N.); ncribeiro@sporting.pt (N.R.); nmloureiro@sporting.pt (N.L.); jparaujo@sporting.pt (J.P.A.); fstavares@sporting.pt (F.T.); 3Centre for Research, Education, Innovation and Intervention in Sport, Faculty of Sport, University of Porto, Rua Dr Plácido Costa 91, 4200-450 Porto, Portugal; jneves@fade.up.pt; 4Tumour & Microenvironment Interactions Group, INEB-Institute of Biomedical Engineering, i3S-Instituto de Investigação e Inovação em Saúde, Universidade do Porto, Rua Alfredo Allen 208, 4200-153 Porto, Portugal

**Keywords:** nutrition, women, high metabolic load distance, internal load, team sports

## Abstract

Football (soccer) is a high-intensity intermittent sport with large energy demands. In a repeated-measures design, we analysed the nutritional intake and training load of fourteen female football players (22.50 ± 4.38 y; 57.23 ± 8.61 kg; 164 ± 6.00 cm; 18.33 ± 2.48% of fat mass and 23.71 ± 2.51 kg of muscle mass) competing in the highest female Football Portuguese League across a typical mid-season microcycle. The microcycle had one match day (MD), one recovery session (two days after the MD, MD+2), three training sessions (MD-3, MD-2, MD-1) and two rest days (MD+1). Energy intake and CHO (g.kg.BW^−1^) intake were lower on the days before the competition (MD+2, MD-3, MD-2 and MD-1 vs. MD; *p* < 0.05; ES: 0.60–1.30). Total distance, distance covered at high-speed running (HSRD) and the high metabolic distance load (HMLD) were lower on MD+2, MD-3 and MD-1 compared with MD (*p* < 0.05; ES: <0.2–5.70). The internal training load was lower in all training sessions before the competition (MD+2, MD-3, MD-2 and MD-1 vs. MD; *p* ≤ 0.01; ES: 1.28–5.47). Despite the small sample size and a single assessment in time, the results suggest that caloric and CHO intake were below the recommendations and were not structured based on the physical requirements for training sessions or match days.

## 1. Introduction

Football (soccer) is a strength- and power-based contact sport involving high-intensity activities intercalated by periods of submaximal activities during ~90 min [1]. Global positioning system (GPS) technology is extensively used during football trainings and competition, providing valid and valuable activity profiles to identify specific physical and physiological demands, field position determinants, individual player-team interaction and accumulated team and player mechanical load [2,3]. Over the years, most of this information has been reported for male football players [2,4,5,6], and little is known of female football players. The physiological (e.g., hormonal milieu) and morphological (e.g., body composition) gender differences become more evident when responding to various training/competition requirements [7,8]. For example, in match play, females covered less distance but at higher intensity levels (maximum speed greater than 15 km.h^−1^) than male players [9], even though most of the match time was spent on low-intensity activities, such as standing, walking and jogging [10,11].

The nature of training, match playing and periods of training and recovery significantly contribute distinct energy and macronutrient requirements [12,13,14]. A robust body of evidence shows that diet is fundamental for the physiological adaptation, recovery and performance in football players [14,15]. In football, the physiological demands rely on both aerobic and anaerobic systems of energy production, which likely lead to the reduction of muscle glycogen stores [16,17]. Nutritional approaches to ensuring adequate glycogen stores to meet the energy costs of training and/or competition are highly endorsed for improving performance, delaying the onset of fatigue, reducing injury risk and supporting physiological adaptation and recovery [15,18,19]. In fact, adequate carbohydrate (CHO) consumption has been recommended [15] (i.e., 5–7 g.kg.BW^−1^ for low- to moderate-intensity training sessions and 7–12 g.kg.BW^−1^ for high-intensity training sessions) for programming the training and recovery days during in-season weeks considering both individual needs and gender [8,15,20,21]. The currently available information focuses largely on male football players [18,19], but a recent systematic review included 20 studies with 462 female field-based team sports athletes and reported that energy and CHO intake per day were insufficient compared with recommendations [22]. 

In football, the microcycle is a structured weekly training unit that is typically labelled according to the number of days following or prior to the match, i.e., match day (MD) [23]. For example, MD-1 refers to one day before MD. Variations in stimuli and training load across the in-season microcycle are important not only for training adaptations but also for appropriate recovery from previous matches and for optimising preparation for upcoming matches, particularly among elite female football players [23]. The majority of studies have failed to contextualize the microcycle by providing specific details of training, recovery or rest days, or mid-season phases [22]. Nutrition should be adapted to individual and collective requirements but also to the intrinsic demands of the weekly schedule of training, recovery, rest days and MDs. To the best of our knowledge, no study evaluated those parameters in elite female football players. Therefore, we aimed to evaluate the training load and self-reported nutritional intake of female football players from the highest Female Football Portuguese League across the in-season microcycle. We hypothesized that self-reported nutritional intake was below the recommendations and was not programmed to the individual physical demands imposed by weekly schedule training.

## 2. Material and Methods

### 2.1. Subjects

From a professional female football club competing in the highest female Football Portuguese League, twenty players were recruited, of whom fourteen completed the study (mean ± SD of the final sample: age, 22.50 ± 4.38 years; body weight [BW], 57.23 ± 8.61 kg; stature, 164 ± 6.00 cm; % body fat, 18.33 ± 2.48; and muscle mass, 23.71 ± 2.51 kg). The participants had 14 ± 4.26 years registered in the football federation and played 4.79 ± 4.47 years as professional players. This team was convenience sampled as one of the few professional female football teams in Portugal and for the ease of researcher access to nutritional and training/competition data. The six players who submitted incomplete dietary records and those with injuries/long-term injuries were excluded from the present study (Figure 1). 

All players were informed about the research protocol, requisites, benefits and risks, and their written consent was obtained before the study began. The study was conducted according to the Declaration of Helsinki (revised version of 2013 at the 64th WMA General Assembly, Fortaleza, Brazil). Since data were obtained as a condition of the players’ employment, they were assessed daily, and therefore, no authorization was required from an institutional ethics committee, and this study is officially considered relieved from institutional approval [24]. 

### 2.2. Study Design 

In a repeated-measures design, 14 of the 20 players included in the present study reported their daily nutritional intake across the one-week mid-season competitive phase in November 2021. We considered a typical week during competition season with one match: a match day (MD) at day six, followed by a day off (MD+1), one regenerative training session (MD+2), and three training sessions before the MD (MD-3, MD-2, MD-1). Nutritional intake and training load were assessed on all days except the rest days (MD+1), on which only nutritional intake was registered (Figure 2). 

### 2.3. Microcycle Contextualization

The training sessions were composed of integrated tactical, technical and physical demands. The MD+2 training session is the first after the MD and has two main goals: (i) improve recovery for the players who completed ≥60 min of competition and (ii) replicate the match mechanical loads for players who completed <60 min on MD (compensatory training). The players who completed ≥60 min performed low-impact regeneration activities. The players who performed <60 min of play practiced high-speed running using high-intensity interval training with short intervals (up to 20 s), followed by a positional match, followed by small-sided games (SSGs) with goalkeepers (area: 30–60 m^−2^ per player). The MD-3 session focused on developing strength and power using positional games and SSGs with goalkeepers (area: 25–50 m^−2^ per player); after the field-based session, the players performed a gym session to develop lower limb strength and power. The MD-2 work aimed to tactically prepare the team for the next match and was characterized by a positional game at a moderate intensity (area: 70–100 m^−2^), an 11 × 11 (72 × 65 m^−2^) match; and strength and power exercises in the gym for the upper limbs. The MD-1 session used activation exercises replicating some technical-tactical elements and ended with set-piece training.

### 2.4. Anthropometric Measures

Anthropometric measurements were taken in the morning after 12 h of fasting. Athletes were assessed in their underwear without footwear and socks. Their height and body mass were measured by means accurate to within 0.1 cm and 0.1 kg, respectively, and were computed based on the arithmetic average of the measurements. Height was measured with an anthropometer (SECA 206) to 0.1 cm accuracy, and body mass was measured with an electronic scale (TANITA BC-601) to 0.1 kg accuracy.

Eight skinfolds (triceps, subscapular, biceps, suprailiac, abdominal, supraspinal, thigh and calf) were assessed twice (at 0.1 mm) with a Harpenden skinfold calliper (British Indicators, Ltd., London, UK). This assessment was performed by a level I-accredited and experienced technician who was blinded to nutritional logs following the International Society for the Advancement of Kinanthropometry recommendations (Stewart et al., 2011). For each athlete, double anthropometric measurements were obtained, and the mean of the two measures was used to calculate the sum of the eight skinfolds (8SKF). An intra-observer technical error of the measurement of 5% for SKF was considered; otherwise, a third measurement was taken. We estimated the total muscle mass using Lee’s equation [25] and the percentage of body fat using Evans’s equation [26], both valid for female athletes.

### 2.5. Nutritional Intake

Daily dietary intake was self-recorded by the football players for seven consecutive days using a food record (foods and beverages consumed over 24 h following the normal team training schedule). The players were instructed on how to fill in the food record, with the aim of minimizing the risk of improperly completed nutrition logs. In addition, they were asked not to alter their usual dietary behaviour during this period. Two registered dietitians were in permanent contact with the athletes to answer any additional questions. The participants self-reported all the foods and fluids they consumed and the relevant information about the times, types and portion sizes of their meals and beverages. The amounts of all food/beverage items were reported in g or portions according to packaging details, and if this information was not available, household measures were recorded. If the answers provided in the assigned 7-day dietary food records were unclear, the athletes were asked to answer further questions on the spot to ensure the maximum accuracy of product consumption [27,28]. The players were also instructed to take pictures whenever they could not assess quantities or provide a proper description. Those pictures were then analysed to complete the records. 

All food records were checked and coded by the same trained nutritionist to ensure equal 7-day food record data throughout the study [29]. From this analysis, detailed information on calories, proteins, carbohydrates (CHO) and lipids consumed was estimated using the Portuguese Food Table or the nutritional information on the package. When it was not possible to obtain this information this way, others nutritional food databases were used, such as TACO or the US nutrition data (https://www.cfn.org.br/wp_content/uploads/2017/03/taco_4_edicao_ampliada_e_revisada.pdf or https://fdc.nal.usda.gov/) (accessed on 10 January 2022)

### 2.6. Training and Match Load

The training load variables over the 7-day period included one match, three training sessions (MD-3, MD-2, MD-1), one regenerative training session (MD+2) and two rest days (the two MD+1). The external training load and match activity profiles of each player were monitored using a portable 18-Hz global positioning system (GPS) device (Apex Pro Series Pod, STATSports, Belfast, UK) incorporated into each player’s jersey on the upper thoracic spine between the scapulae. In order to decrease the variability across different devices, each player used the same device during the study period [30]. The GPS units were turned on before the warm-up and turned off after the completion of the training sessions. Satellite data sampled at 18 Hz provided measures of total distance (m) covered and distance covered at specific velocity bands ≥4.4 m.s^−1^ and ≥5.6 m.s^−1^ corresponding to H-SRD and sprint distance, respectively. The used thresholds are typically used in female-soccer specific literature [31].

The high metabolic load distance (HMLD), a metric frequently utilised to determine training load in female football [32], was also quantified as the total amount of HSRD and the total distance of accelerations and decelerations ≥2 m.s^−2^ throughout a training session or a match. HMLD represents the distance (m) covered by a player when their metabolic power exceeded 25.5 W.kg^−1^.

### 2.7. Internal Training and Match Load 

The proxy of internal training load (iTL) was calculated as the product of the individual session rate of perceived exertion (RPE) and the duration of the session/match [33]. This parameter is a simple, non-invasive and inexpensive method for monitoring training load [34]. Furthermore, it is considered valid and reliable for female field-based sports, including football [34,35]. The individual training RPE of each training session or match was obtained ~30 min after the completion of the session to ensure that each player reported a global RPE for the entire session rather than the most recent exercise [33,36]. Each player was confidentially interviewed and not allowed to see the other players’ values. Before the study, the CR10 scale was explained to players verbally and used for two weeks to ensure that they were familiarized with it.

### 2.8. Statistical Analysis 

The normality of the variables’ distribution was tested using the Shapiro-Wilk test. Mean ± standard deviation (SD) or median and interquartile range (IQR) were used to present descriptive statistics as the data presented nonnormal distributions. The nonparametric Wilcoxon’s signed-rank test was used to analyse the differences between the days. Effect sizes (ES) were calculated to determine meaningful differences and were classified as follows: trivial (<0.2), small (0.2–0.6), moderate (>0.6–1.2), large (>1.2–2.0) and very large (>2.0–4.0) [37]. Spearman’s correlations were performed to analyse the correlation between nutritional and training load variables and were classified as follows: negligible (0.0–0.1), weak (0.10–0.39), moderate (0.4–0.69), strong (0.70–0.89) and very strong (0.90–1.00) [38]. All analyses were conducted using Statistical Package for Social Sciences software program (SPSS, version 27; IBM, Armonk, NY, USA), and the alpha level was set a priori at 0.05.

## 3. Results

The mean self-reported daily energy intake was 1764.37 ± 495 kcal, which corresponds to an average of 38.93 ± 13.22 kcal.kg.free fat mass.day^−1^. The average % of caloric intake was 48.14 ± 8.10 for CHO, 23.30 ± 5.36 for protein and 27.73 ± 8.05 for lipids. Energy intake was lower on MD+1 (*p* = 0.05; ES: 1.29), MD+2 (*p* = 0.009, ES: 1.12), MD-3 (*p* = 0.049; ES: 0.99), MD-2 (*p* = 0.038; 0.96); and MD-1 (*p* = 0.01, ES:1.30) vs. MD, denoting moderate to large effects. Moderate to large ES were also observed for relative CHO (g.kg.BW^−1^) intake, which was lower on day #1 MD+1 (*p* = 0.05, ES: 0.60), #7 MD+1 (*p* = 0.026, ES: 1.21), MD+2 (*p* = 0.03, ES: 0.73), MD-3 (*p* = 0.041, ES: 0.70) and MD-1 (*p* = 0.048, ES: 0.71) compared with MD. The relative CHO intake was lower on day #7 MD+1 (*p* = 0.02, ES: 0.82) vs. MD-2, again with a moderate ES. No significant differences were observed in relative protein or lipid consumption (Figure 3). 

Total distance was lower in MD+2 (*p* = 0.003; ES: 1.79), MD-3 (*p* = 0.003; ES: 3.68) and MD-1 (*p* = 0.002; ES: 5.70) vs. MD, whereas on MD-2, the total distance covered was higher than that on MD (*p* = 0.05, ES: 1.38), with large to very large ES. Total distance on MD+2 (*p* = 0.025; ES: 1.78, large effect), MD-3 (*p* = 0.012; ES: 2.73, very large effect) and MD-1 (*p* = 0.017; ES: 2.36, very large effect) was lower than on MD-2. H-SRD was lower on MD+2 (*p* = 0.003; ES: 0.70), MD-3 (*p* = 0.003; ES: 2.08), MD-2 (*p* = 0.01; ES: 1.67) and MD-1 (*p* = 0.002; ES: 2.28) vs. MD, denoting ES ranging from small (e.g., MD-4) to very large (e.g., MD-1). Moreover, H-SRD was lower on MD-3 (*p* = 0.003; ES: 1.99), MD-2 (*p* = 0.003; ES: 1.38) and MD-1 (*p* = 0.003; ES: 2.25) vs. MD+2. HMDL was lower on MD+2 (*p* = 0.004; ES < 0.2, trivial effect), MD-3 (*p* = 0.003; ES: 1.47, large effect) and MD-1 (*p* = 0.002; ES: 1.98, large effect) vs. MD. HMDL was lower on MD-3 (*p* = 0.003; ES: 1.85, large effect) and MD-1 (*p* = 0.003; ES: 2.86, very large effect) vs. MD+2. The iTL was lower on MD+2 (*p* = 0.006, ES: 2.51, very large effect), MD-3 (*p* = 0.001, ES: 5.47, very large effect) and MD-1 (*p* < 0.001, ES: 4.40, very large effect) vs. MD (Figure 4). The iTL was lower on MD-3 (*p* = 0.001, ES: 5.11) and MD-1 (*p* = 0.001, ES: 3.42) compared with MD+2 and with very large ES in both cases. 

Furthermore, moderate correlations were found between self-reported energy intake and total distance (*p* = 0.043; r = 0.618) on MD-3 and between energy intake and iTL (*p* = 0.035; r = 0.566) and CHO intake and iTL (*p* = 0.047; r = 0.539) on MD. Overall, for most training days, no correlation was found between the self-reported nutritional intake variables and the training load variables.

## 4. Discussion

Considering the relevance of proper nutrition for professional female athletes’ performance and health, the efforts to understand macronutrients intake in female’s athlete are well warranted [22,39]. To meet the energy requirements of a high-intensity training schedule, a well-structured diet with adequate energy and macronutrient amounts provides the nutritional needs for peak physical and physiological demands [39]. In female football players, the information regarding nutritional intake and training load is scarce, and the majority of the available studies did not contextualize the microcycle, i.e., provide specific details on training, recovery or rest days or mid-season phase [22]. Therefore, we aimed to evaluate the self-reported nutritional intake and training load among female football players in the elite Female Football Portuguese League across the in-season microcycle for better understanding their correlations and variations across a typical mid-season microcycle. Our main findings demonstrated that (i) the athletes’ self-reported nutritional intake was below the recommendations (throughout all days of the selected microcycle), (ii) the training load was generally lower on all days compared with MD, and (iii) high-intensity training load sessions and regenerative training sessions did not correlate with the highest self-reported nutritional intake, contributing novel findings to the knowledge on this topic.

The self-report energy intake in the present female team fell short of the recommendations proposed by Dobrowolski et al. [21], being <1115 kcal and <490 kcal below the recommendations on in training (mean ± SD, 2797.13 ± 65.55 kcal) and rest (2276.56 ± 88.28 kcal) days, respectively. Despite the small sample size, our findings agree with studies reporting that female football players did not achieve the recommendations for energy intake [13,40,41]. Data from the present study suggest that the football players comply with macronutrient recommendations [21] except for CHO, which was slightly below the recommendations and was also lower on all days in the microcycle in comparison with MD, suggesting that CHO intake to meet physical demands of the training and recovery days has not been properly structured for this team. Considering that the physiological demands of football rely on both aerobic and anaerobic systems of energy production [16,17], proper CHO intake before competition to increase glycogen stores to meet the energy costs of training sessions and during competition should be part of a nutritional structuration, particularly for female football players [8,15,20]. Although there are current recommendations for adequate amounts of CHO consumption for female football players [20], several studies reported findings similar to ours [40,42,43]. 

It should be noted that although the energy and CHO intake here were below the recommendations, this could have been partly due to a lack of compliance by the players and not entirely because of inadequate programming. The inadequate CHO intake during the high-intensity training days may be also attributed to appetite-regulating hormones [44,45,46], including ghrelin, acylated ghrelin, glucagon-like peptide 1 and YY peptide. The role of these hormones in the suppression of hunger after high-intensity exercise in female athletes is well-known [44,45].

Protein is important for maintaining proper bodily functions and maximizing recovery in order to sustain performance throughout the season [15]. It is well accepted that intake in the magnitude of 1.2 to 2 g.kg.BW^−1^ is enough to meet physical requirements [20]. Although these values were determined for men, there is no reason to believe they should be different for female athletes [7]. In line with previous studies [13,40,43,47,48], we observed that the team’s protein intake met the recommendations [21]. 

Lipids are an important source of energy, fat-soluble vitamins and essential fatty acids [49]. Lipid intake is critical for health and has been described as contributing to the optimal testosterone levels in blood [50]. A low-fat diet may compromise health, as it reduces the absorption of fat-soluble vitamins and glycogen storage in muscle [49,50]. The recommendations for lipid intake were based on the percentage of energy intake [21] and for female football players should be between 20 and 30% of the total energy intake in order to not compromise protein and carbohydrate intake. Data from the present study showed that lipid intake complied with the recommendations. Regarding the energy value of the diet, the guidelines recommended at least 30 kcal.kg.free fat mass.day^−1^ to prevent negative health and performance consequences of low energy availability [51]. In the present study, we observed an average of 38.92 kcal.kg.free fat mass.day^−1^ in training days and 47.2 kcal.kg.free fat mass.day^−1^ in MD. 

HMDL is an estimation of energy cost, based mainly on the movement profile of the athlete and the high-intensity actions performed at high and low displacement speeds [32,52,53]. In the present study, the lowest values for HMDL were found on MD+2, MD-3 and MD-1 (vs. MD) but not on MD-2. However, no correlations were observed between HMDL and the nutritional variables, suggesting room for improving the nutritional programming on this team. HMDL is widely used but is mainly measured in male football players, and of our best knowledge, no study reported this parameter in female football players. Therefore, the present study brings a novel finding by evaluating the HMDL variations throughout a microcycle in female football players and correlating these variations with nutritional intake. 

We further observed that the highest HSRD was on MD+2, which may be explained by the high demands imposed on the players who had completed <60 min in the previous match (*n* = 9). For players who completed ≥60 min during the match (*n* = 5), MD+2 was a regenerative training session, but for the others, MD+2 is a compensatory training day based on high-intensity interval training and SSG. As opposed to previous studies [4,54], we quantified the internal and external training loads during a typical microcycle, characterized by training sessions with distinct goals and demands. For this reason, our findings are discussed in context of a typical mid-season microcycle regarding self-reported nutritional intake and training load in female football players. However, since our data refer to a single microcycle, it is unclear whether the findings truly reflect regularities (which would be expected, being a typical microcycle) or instead idiosyncrasies of the particular week that was analysed.

Some limitations of the present study should be considered when interpreting the findings. Self-reported nutritional intake is likely to be misreported and may not completely reflect participants’ true nutritional intakes (e.g., intake amounts could be under- and/or over-estimated). However, some studies mentioned that seven days is sufficient for assessing energy intake but not macronutrient intake [55], which may require assessment over a longer period of time. In addition, the self-reported nutritional intake over seven days should be analysed with caution as it may not represent a given participant’s long-term nutrient status. Considering one single assessment in time, i.e., one microcycle, and the small sample size, it is unclear whether these findings can be generalized to other teams/players.

## 5. Conclusions

In summary, there was a mismatch between self-reported nutritional intake and training demands, and there was lower self-reported energy intake on all 5 weekdays in comparison with the MD. There was no increase in energy intake on the day after the match, or any attempt to build reserves on the days before the match. Furthermore, training should prepare for the demands of matches, but in this case, players were underloaded through the entire week, never meeting the demands of the match. The absence of correlations between macronutrient intake and training load means that the programming of the microcycle is not appropriately integrating training and nutritional strategies and loads or that athletes are not following the nutritional program. Due to the small sample size and to data representing a single microcycle, it is unclear how generalizable these results are, but they still raise some concerns related to the need to (i) better adjust training demands to prepare to meet match demands and (ii) better adjust training and nutritional programming.

## Figures and Tables

**Figure 1 nutrients-14-02149-f001:**
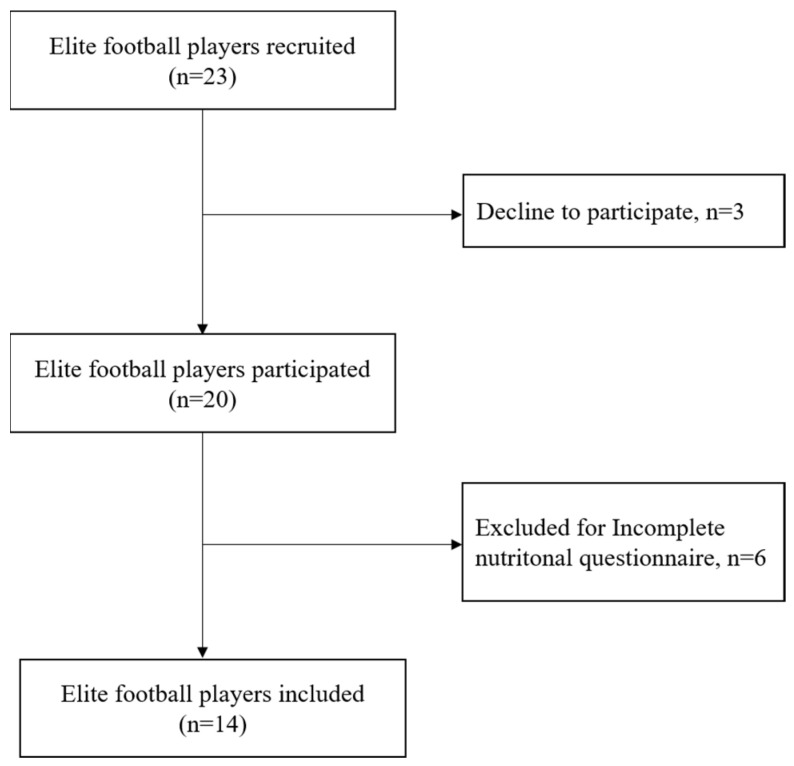
Flow diagram of the participants.

**Figure 2 nutrients-14-02149-f002:**
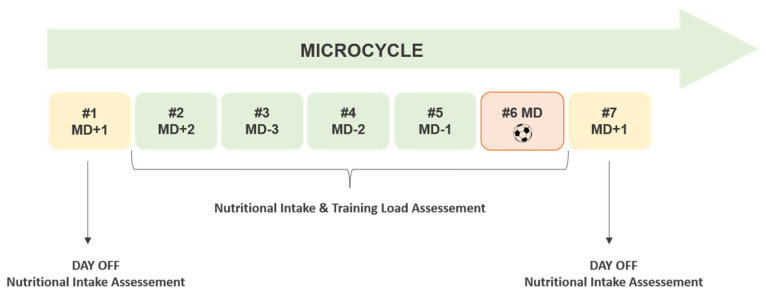
A typical mid-season microcycle during the competitive phase of a women’s football championship.

**Figure 3 nutrients-14-02149-f003:**
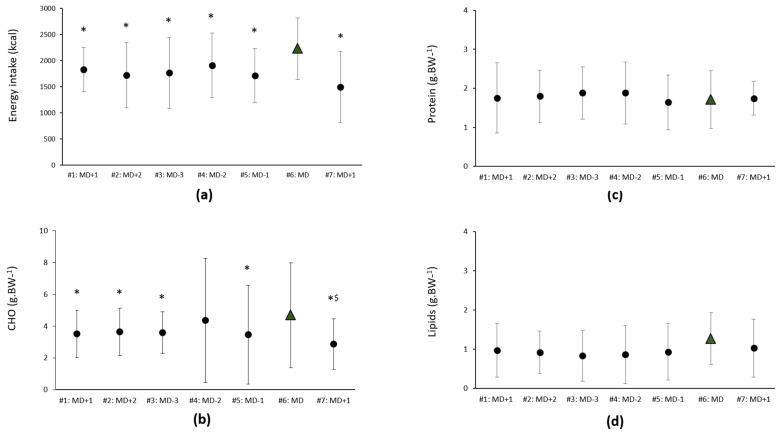
Self-reported energy, CHO, protein and lipids intake across the mid-season microcycle. (**a**) Energy and relative (**b**) CHO, (**c**) protein, (**d**) lipids intake. Data are expressed as median (ITQ); * vs. MD; ^$^ vs. MD-2.

**Figure 4 nutrients-14-02149-f004:**
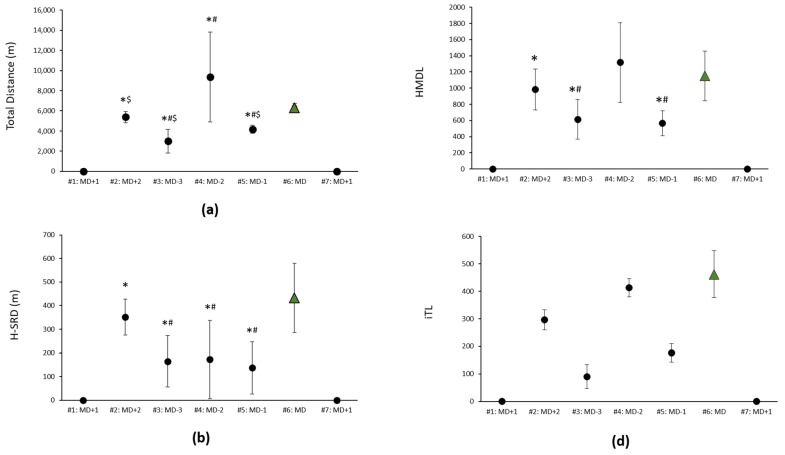
Internal training load and variables of external training load: (**a**) total distance, (**b**) H-SRD, (**c**) HMLD and (**d**) iTL. Data are expressed as median (ITQ); * vs. MD; ^#^ vs. MD+2; ^$^ vs. MD-2.

## Data Availability

Data will be provided to all interested parties upon reasonable request.

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
