# Peer review of "Nutritional Intake and Training Load of Professional Female Football Players during a Mid-Season Microcycle"

_nutrients, 2022, doi:10.3390/nu14102149_

Round 1

Reviewer 1 Report

  • The authors studied the nutritional intake and training load of professional female 2 football players during a mid-season microcycle. The study is very informative and adds value to the current research body; however, some points should be addressed.

Abstract

  • The abstract needs to contain other anthropometric characteristics of participants such as height, weight, etc.
  • The p-value should be mentioned for all the significant changes mentioned in the abstract.

Introduction

  • The study hypothesis needs to be added at the end of “Introduction”.

Methods

  • Day-to-day test reliability, CV range, and intraclass correlation coefficients for the assessments need to be included for ALL the assessments.

Discussion

  • The study novelty needs to be explained in the “Discussion”.
  • The Study limitations part is missing from the manuscript, and there are numerous limitations to be listed.

Author Response

We would like to thank the careful review of our manuscript. It is our belief that your comments and suggestions certainly enriched the quality of the text and improved substantially the rationale of the discussion. All the suggestions were therefore considered and included in the revised manuscript. They are highlighted by "Track Changes" function.

Reviewer #1

The authors studied the nutritional intake and training load of professional female 2 football players during a mid-season microcycle. The study is very informative and adds value to the current research body; however, some points should be addressed.

Abstract: The abstract needs to contain other anthropometric characteristics of participants such as height, weight, etc.

The p-value should be mentioned for all the significant changes mentioned in the abstract.

We completely agree with your comment. Because of restriction on words number, we mentioned the general p-value and range of effect size to have an idea of significance and the magnitude of the differences.

Using the words number (not exceed 200 words) we add weight, height, %of body fat and muscle mass information. Please, see line 22.

Introduction: The study hypothesis needs to be added at the end of “Introduction”.

Modified in accordance. Please, see line 81-83.

Methods: Day-to-day test reliability, CV range, and intraclass correlation coefficients for the assessments need to be included for ALL the assessments.

Day-to-day test reliability does not seem to have the proper application in the present study as we evaluated the variations of training load and nutritional intake thought a typical microcycle in football. So, the day-to-day variations observed might be due to the different and specific physical demands of each training day, which would result in an error of reliability test. In support of this, we described in detail the microcycle in material and methods section, 2.3 microcycle contextualization.

To support our argument we based on Shou Y. et al 2022 [1] that states:  “the test-retest reliability assumes that the true score being measured is the same over a short time interval. To be specific, the relative position of an individual's score in the distribution of the population should be the same over this brief time period”, which is one of the test-retest reliability techniques used: “generalizability theory approach” [2]. Moreover, “the error component of the true score equation comes from the variability of the observed values on occasions at different time point”. In the present study, we aimed to evaluate the variations of nutritional intake and training load through a week, a very short period of time. Considering that the “test-retest reliability can be estimated by calculating the correlation coefficient of the measured values at two separate time points” [2], we should have data of two microcycles, i.e., two typical weeks, in order to measure two separate time points. In this case, it would be appropriate to apply day-to-day reliability. However, we do not have these data and we clearly assume that it is a limitation of our study.  

Reliability is defined as the extent to which measurements can be replicated, i.e., it reflects the degree of correlation and the agreement between measurements [2]. In line with this line and in accordance with aim of our study, we determined the correlation (Spearman) between measurements (line 251) to observe the degree of correlation (please, see results section, line 309), i.e., within nutritional variables and training load variables and between those variables. For example, the caloric intake was correlated with CHO intake, all these degree of correlations and agreement between the variables were determined.

Discussion: The study novelty needs to be explained in the “Discussion”. The Study limitations part is missing from the manuscript, and there are numerous limitations to be listed.

Modified in accordance. Please see lines 338-340, line 388-391.

Regarding limitations, we mentioned them throughout the manuscript. However, to state clearer we added a chapter where the limitations are listed. 

Reviewer 2 Report

Comments to the work entitled "Nutritional Intake and Training Load of Professional Female Football Players During a Mid-Season Microcycle".

  1. Please formulate (and emphasize) the purpose of the work (research problem) more clearly.
  2. In the 'Subjects' section, please provide the players' experience (M, SD, scope).
  3. In the 'Nutritional Intake' section, what computer nutrition program was used to evaluate energy and macronutrient intake? Furthermore, to what nutritional standards were energy and macronutrient consumption related? I also have a question as why the consumption of only basic macronutrients was assessed (indicate this problem in the 'Limitations').
  4. In the 'Results' section, it is worth considering whether, in addition to the average energy supply, the average supply of macronutrients should also be taken into account. Moreover, what was the % of energy from 'Proteins", 'Fats' and 'Carbohydrates'?
  5. In the 'Discussion' section, there are no references to other studies with regard to lipid intake. The 'Discussion' requires some overall edition and structuring according to the order of the results (after the discussion of nutritional results, please discuss the results related to training loads, and then move on to overviewing the relationship between exercise characteristics and nutritional factors). At the end of the 'Discussion', please add a paragraph on 'Limitations of the study'.

Author Response

Please formulate (and emphasize) the purpose of the work (research problem) more clearly.

Modified in accordance.

In the 'Subjects' section, please provide the players' experience (M, SD, scope).

We totally agree with your suggestion. We added this information (please, see lines 94-95).

In the 'Nutritional Intake' section, what computer nutrition program was used to evaluate energy and macronutrient intake? Furthermore, to what nutritional standards were energy and macronutrient consumption related? I also have a question as why the consumption of only basic macronutrients was assessed (indicate this problem in the 'Limitations').

The software used was nutritics (NUTRITICS, 2019. RESEARCH EDITION (V5.096), DUBLIN).

The recommendations values used in the present study were based on Dobrowolski et al 2020 “Nutrition for Female Soccer Players-Recommendations” Medicina (Kaunas),  56 (1), and not Dobrowolski and Wlodarek 2019 “Dietary Intake of Polish Female Soccer Players” Int J Environ Res Public Health 16  (7) as initially mentioned.  It was an error and now it is corrected.

The assessment of macronutrients was defined as the main aim of the present study. As we aimed to evaluate the training load and the nutritional intake and, therefore, understand whether nutritional intake was concordant with training load during a one typical week of training. For this reason, the energy, CHO, protein and lipids intake in football was considered in the present study.

The micronutrients assessment is not within the scope of our study and considering their relevance should be addressed in future studies. For this reason, we consider that it is not a limitation.

In the 'Results' section, it is worth considering whether, in addition to the average energy supply, the average supply of macronutrients should also be taken into account. Moreover, what was the % of energy from 'Proteins", 'Fats' and 'Carbohydrates'?

We understand your suggestion, and we also calculated the % of energy derived from CHO, protein and lipids. We added in the results section the following information:

The average of % of caloric intake was 48.14±8.10 for CHO, 23.30±5.36 for protein and 27.73±8.05for lipids.

  1. In the 'Discussion' section, there are no references to other studies with regard to lipid intake. The 'Discussion' requires some overall edition and structuring according to the order of the results (after the discussion of nutritional results, please discuss the results related to training loads, and then move on to overviewing the relationship between exercise characteristics and nutritional factors). At the end of the 'Discussion', please add a paragraph on 'Limitations of the study'.

We totally agree. Modified in accordance.

Some sentences were added to support the information previously described, as reviewer #2 suggested.
